# Consumption of Lactose, Other FODMAPs and Diarrhoea during Adjuvant 5-Fluorouracil Chemotherapy for Colorectal Cancer

**DOI:** 10.3390/nu12020407

**Published:** 2020-02-04

**Authors:** Reetta Holma, Reijo Laatikainen, Helena Orell, Heikki Joensuu, Katri Peuhkuri, Tuija Poussa, Riitta Korpela, Pia Österlund

**Affiliations:** 1Faculty of Medicine, Pharmacology, Medical Nutrition Physiology and Human Microbe Research Program, University of Helsinki, P.O. Box 63, FI-00014 Helsinki, Finland; reetta.holma@gmail.com (R.H.); reijo.laatikainen@booston.fi (R.L.); riitta.korpela@helsinki.fi (R.K.); 2Booston Oy Ltd., Viikinkaari 6, FI-00790 Helsinki, Finland; 3Department of Clinical Nutrition Therapy, Helsinki University Hospital, P.O. Box 100, FI-00029 HUS, Finland; helena.orell@hus.fi; 4Department of Oncology, Helsinki University Hospital and University of Helsinki, P.O. Box 180, FI-00029 HUS, Finland; heikki.joensuu@helsinki.fi; 5Social Services and Health Care Division, P.O. Box 6230, FI-00099 City of Helsinki, Finland; katri.peuhkuri@hel.fi; 6STAT-Consulting, Vahverokatu 6, FI-37130 Nokia, Finland; tpoussa@netti.fi; 7Department of Oncology, Tampere University Hospital and Tampere university, P.O. Box 2000, FI-33521 Tampere, Finland

**Keywords:** chemotherapy, colorectal cancer, diarrhoea, 5-fluorouracil, FODMAP, gastrointestinal symptoms, hypolactasia, lactose intolerance

## Abstract

Chemotherapy-induced mucosal injury of the small intestine may interfere with the enzymes and transporters responsible for the hydrolysis and absorption of dietary carbohydrates causing diarrhoea, abdominal discomfort and pain. The aim of this study was to investigate the association between the consumption of foods rich in FODMAPs (fermentable oligo-, di- and monosaccharides and polyols) and gastrointestinal symptoms in patients receiving adjuvant therapy for colorectal cancer. The patients (*n* = 52) filled in a 4-day food diary at baseline and during therapy and kept a symptom diary. The intakes of FODMAP-rich foods were calculated as portions and the intakes were divided into two consumption categories. Patients with high consumption of FODMAP-rich foods had diarrhoea more frequently than those with low consumption (for lactose-rich foods the odds ratio (OR) was 2.63, *P* = 0.03; and for other FODMAP-rich foods 1.82, *P* = 0.20). Patients with high consumption of both lactose-rich and other FODMAP-rich foods had an over 4-fold risk of developing diarrhoea as compared to those with low consumption of both (OR, 4.18; *P* = 0.02). These results were confirmed in multivariate models. Conclusion: Consumption of lactose-rich foods results in an increased risk of diarrhoea during adjuvant therapy for colorectal cancer, especially when the consumption of other FODMAP-rich foods is also high.

## 1. Introduction

Cytotoxic chemotherapy may damage the gastrointestinal tract mucosa and cause mucosal inflammation, oedema, ulceration and tissue atrophy. Chemotherapy-associated gastrointestinal toxicity often manifests as diarrhoea, abdominal discomfort and pain; it may also lead to treatment interruptions and even discontinuation [1,2]. A mucosal injury in the brush border of the small intestine can interfere with the function of the enzymes and transporters that are responsible for hydrolysis and absorption of dietary carbohydrates [3]. Unabsorbed carbohydrates end up in the colon, where they are fermented by microbes to hydrogen, carbon dioxide and short chain fatty acids, which, when abundant, act as osmotic laxatives and cause symptoms such as diarrhoea, flatulence and distension of the abdomen [4,5].

Adjuvant 5-fluorouracil chemotherapy for colorectal cancer may be associated with reversible hypolactasia and secondary lactose intolerance [6] but the efficacy of lactose-restriction in reducing gastrointestinal toxicity during chemotherapy is unknown. Similarly, the effects of malabsorption of other carbohydrates, such as fructose, on the symptoms evoked by chemotherapy remain to be clarified. The fructose absorption capacity of the small intestinal epithelial cells is limited but can be considerably enhanced by co-ingestion of glucose or galactose [4,7]. Fructose is increasingly found in the Western diet [7].

A specific diet developed to alleviate symptoms of patients with the irritable bowel syndrome, called the FODMAP-restriction diet, aims to reduce the intake of poorly absorbed (fermentable) oligo-, di- and monosaccharides and polyols (FODMAPs), specifically fructans (inulin, fructo-oligosaccharides), galacto-oligosaccharides, raffinose, lactose, fructose and polyols (sorbitol, mannitol, maltitol, xylitol and lactitol) [8]. Lactose is considered as a FODMAP for patients with hypolactasia. FODMAPs are quickly fermented in the proximal colon and are thought to be the principal cause of food-related gastrointestinal symptoms in functional gastrointestinal disorders. A diet low in FODMAPs is an effective dietary means to reduce these kinds of symptoms in these patients [9].

It is well-known that chemotherapy and radiotherapy exert toxic effects on the intestinal mucosa [10]. Indeed, a non-controlled pilot-study demonstrated that a low FODMAP diet alleviated symptoms and improved the quality of life in cancer patients with radiation-induced acute intestinal injury [11], i.e., further evidence for a link between a high FODMAP intake and cancer-therapy-induced diarrhoea, thus encouraging us to examine if a low FODMAP diet could also benefit patients receiving chemotherapy.

The extent to which lactose and other poorly absorbed short chain carbohydrates contribute to the gastrointestinal adverse effects evoked by 5-fluorouracil, the cornerstone for gastrointestinal cancer treatments in both the adjuvant and metastatic setting, or other chemotherapeutic agents is unknown. The objective of this study was to investigate whether the consumption of foods rich in FODMAPs would be associated with gastrointestinal symptoms during adjuvant therapy for colorectal cancer.

## 2. Materials and Methods 

### 2.1. Subjects and Study Design 

The study was performed within the context of a prospective randomised factorial design study (the LIPSYT study, ISRCTN98405441). The LIPSYT study focused on adverse events of chemotherapy regimens that were given after surgery for colorectal cancer, chemotherapy-related secondary lactose intolerance and involving a *Lactobacillus rhamnosus* GG intervention in an attempt to reduce chemotherapy-associated adverse effects [6,12].

Briefly, the LIPSYT study compared two adjuvant 5-fluorouracil-based chemotherapy regimens, the Mayo regimen and the simplified de Gramont regimen, as adjuvant treatments for colorectal cancer and further assessed the effects of a probiotic *Lactobacillus rhamnosus* GG with or without guar gum fibre on the adverse effects of chemotherapy [12] (Appendix A). Briefly, the patients were randomized by applying an open-label, 2 (chemotherapy) × 3 (supplements) factorial study design, to receive adjuvant chemotherapies, the simplified de Gramont or the Mayo regimen and supplements, *Lactobacillus rhamnosus* GG capsules (ATCC 53103, Valio Ltd., Helsinki, Finland), *Lactobacillus rhamnosus* GG with guar gum fibre, or no supplements for 24 weeks. The minimization method with gender, tumour site and stage as prognostic factors were used to ensure that an equal number of participants were allocated to the study groups.

Fifty-two (43%) out of the 120 patients who participated in the randomised adjuvant nutritional intervention study could be included in the current FODMAP study (Figure 1). 

Patients were referred to the Helsinki University Hospital, Department of Oncology, for adjuvant treatment of colorectal carcinoma. Inclusion criteria comprised age between 18 and 75 years and histologically confirmed colorectal adenocarcinoma that was radically removed at surgery and rendered free from all overt metastases. All patients had undergone one of the following surgical procedures: right-sided hemicolectomy; left-sided hemicolectomy, Hartmann or sigma resection; anterior or abdominoperineal resection; and total or subtotal colectomy (Table 1). All patients started a 24-week adjuvant, 5-fluorouracil-based chemotherapy course. After the randomization, the groups consisted either of the Mayo regimen (intravenous leucovorin 20 mg/m^2^ and a 3–5 min intravenous bolus of 5-fluorouracil, 370 to 425 mg/m^2^, on days 1–5 of the cycle, repeated six times at 4-week intervals) or the simplified de Gramont regimen (a 2-h infusion of leucovorin 400 mg/m^2^, followed by a bolus of 400 mg/m^2^ 5-fluorouracil bolus and a 46-h infusion of 3.03.6 g/m^2^ 5-fluorouracil, repeated at 14-day intervals during 24 weeks) [12]. Patients with rectal cancer (*n* = 20) received pelvic radiotherapy, either preoperative radiotherapy (*n* = 6) or postoperative chemoradiation (*n* = 14). Preoperative radiotherapy consisted of five 5-Gy fractions on weekdays. Postoperative chemoradiation was given to a cumulative dose of 50.4 Gy in 1.8 Gy daily fractions for 5.5 weeks. The doses of 5-fluorouracil were reduced during the cycles given during pelvic radiotherapy (cycles III and IV). The study protocol was approved by the Institutional Review Board of Helsinki University Hospital and conforms to the provisions of the Declaration of Helsinki. The trial was registered (identifier ISRCTN98405441). The study participants provided written informed consent prior to initiation of the treatments. 

We required that each FODMAP study participant kept a reliable food diary throughout the study, underwent at least 3 months of chemotherapy and did not have metastatic disease that might interfere with the evaluation of symptoms. A comparison of the baseline characteristics of the LIPSYT trial patients [12] who were included in the FODMAP study and those who were excluded is provided in Table 1. 

The patients filled in a 4-day food diary both before the beginning of the first adjuvant chemotherapy cycle (baseline) and during the third chemotherapy cycle and kept a symptom diary of adverse effects, including gastrointestinal complaints during the first and third month of adjuvant chemotherapy administration (Appendix A). The reasons for choosing the third therapy cycle as the assessment point for symptoms and dietary intake were (1) only minor dose reductions were performed after cycles I-II, (2) patients had received the treatment for 9–12 weeks, (3) dietary intake was not affected by the results of the secondary lactose tolerance test (which was given to participants after completion of the third chemotherapy cycle on week 12 and (4) secondary lactose intolerance had stabilized and had been assessed.

### 2.2. Concurrent Medications, Assessment of Methane Production and Oral Lactose Tolerance Test

All necessary medication for appropriate treatment of concurrent diseases and treatment associated adverse effects (for example, loperamide or octreotide for diarrhoea, antibiotics for infection, metoclopramide and dexamethasone and granisetron/ondansetron/tropisetron as antiemetics) were permitted. 

Colonic methane production was measured before chemotherapy/intervention (at baseline) as previously described [13]. Briefly, the categorization of subjects into methane producers and non-producers was based on in vitro faecal fermentation results or on a breath test. No antibiotics, enemas or laxatives were used for at least two weeks prior to sampling. 

Lactose tolerance was evaluated at the baseline and after 12 weeks of adjuvant therapy (in the middle of the therapy) by administering 50 g of lactose after a 12-h fast as previously described [6]. Blood glucose levels were measured at 20-min intervals for 40 min (hypolactasia, a blood glucose level increase <1.1 mmol/L; borderline lactase deficiency, an increase of 1.1–1.6 mmol/L; no lactase deficiency, an increase >1.6 mmol/L). The patients were informed about the results of the baseline lactose test after completion of the first cycle at 4 weeks. Those with lactase deficiency at baseline were instructed to consume either lactose-free or low-lactose foods from the beginning of the second cycle onwards during the whole study. A second lactose tolerance test was carried out after the third month of chemotherapy at 12 weeks; the results were provided to the patients when they had filled in their food diaries and symptom diaries for the 3rd cycle.

### 2.3. Analysis of Food Diaries

Food diaries were analysed by a trained nutritionist (RH, who was blinded to other clinical information) for the intake of lactose-rich foods and foods rich in other poorly absorbed short chain carbohydrates i.e., fermentable oligo-, di- and monosaccharides and polyols (FODMAPs) [14,15]. Lactose-rich foods included in the analysis were liquid milk products (milk, buttermilk, yogurt, cream), soured whole milk, curd, ice cream and porridge prepared in milk. Low-lactose or lactose-free milk products were not included in the estimates. Lactose-rich foods were calculated in decilitres (dL). FODMAP-rich food (calculated as portions) included in the estimate were breads, baked confectionary such as buns, cakes and biscuits (1 portion corresponds to 1 piece), porridges prepared from wheat, rye and barley (common dishes in Finland) (1 portion corresponds to 2 dL), muesli (1 portion corresponds to 1 dL), apples, pears, nectarines, apricots, peaches, plums (1 portion corresponds to 1 fruit), watermelon, cherries, cauliflower, cabbage, peas, (1 portion corresponds to 1 dL), apple juice and pear juice (1 portion corresponds to 2 dL). As far as we are aware, there is no computer software for the analysis of the FODMAP content (other than lactose and fructose) of Finnish foods. Therefore, the intakes of lactose-rich foods were calculated as decilitres and FODMAP-rich foods were calculated using a portion as a measuring unit, thus considering the known and reliably self-reported FODMAPs. FODMAP-rich foods were derived from the given references [12,13]. The data were categorized in two or three consumption categories based on the intake of portions of FODMAPs or lactose rich foods measured in decilitres per day.

### 2.4. Assessment of Gastrointestinal Symptoms

The patients kept a diary of their gastrointestinal symptoms during chemotherapy and chemotherapy-related adverse events were classified and graded every four weeks by an oncologist (PÖ) according to the Common Toxicity Criteria of the National Cancer Institute of Canada scale, version 2. The adverse events were graded according to the highest grade recorded during the first month of adjuvant therapy administration (one 28-day cycle of the Mayo or two 14-day cycles of de Gramont regimen) and according to the highest grade recorded during the third month of adjuvant therapy (i.e., two periods). Thus, all patients included in the analysis received at least half of the planned chemotherapy cycles.

### 2.5. Statistical Analysis

Chemo(radio)therapy related gastrointestinal symptoms (diarrhoea, constipation, vomiting, flatulence, dyspepsia and stomatitis) of the highest intensity on a scale from grade 1 (mild) to grade 4 (life-threatening) were the primary variables examined in this study. The consumptions of lactose-rich (dL/day) and FODMAP-rich foods (portions/day) were the primary explanatory variables. They were considered as ordered data and were categorized according to median (above median consumption (High) vs. below median consumption (Low)) and according to tertiles (highest, moderate vs. lowest consumption). The analysis detected an interaction between lactose-rich (High vs. Low) and FODMAP-rich (High vs. Low) consumptions in diarrhoea and flatulence. Therefore, the consumptions of lactose-rich and FODMAP-rich foods were combined into one composite variable with four categories (Low–Low, High–Low, Low–High and High–High). Two periods of both the gastrointestinal symptoms (highest grade) and of the consumption of lactose-rich and FODMAP-rich foods were considered during the FODMAP study. The association between the consumption of lactose-rich and FODMAP-rich foods at the baseline vs. gastrointestinal symptoms during the first cycle and the same association during the third cycle were not analysed separately. Instead, the associations between consumptions of lactose-rich and FODMAP-rich foods vs. gastrointestinal symptoms were assessed using the generalized estimating equations (GEE) analysis with binary logistic regression. The results are given as odds ratios (OR, with a 95% confidence interval). The final models were adjusted for covariates. Treatment (simplified de Gramont vs. Mayo regimen), *Lactobacillus rhmannosus* GG intervention (yes vs. no), fibre supplement (yes vs. no), baseline lactose intolerance (yes vs. no), stoma (yes vs. no), methane producer (yes vs. no) and postoperative radiotherapy (yes vs. no) were utilized as categorical covariates if their *P*-value was <0.20 in the univariate analysis. Covariates were excluded one by one from the model if their *P*-value was ≥0.20 after their introduction into the model. The GEE analysis with ANOVA was applied when the consumption of lactose-rich foods during cycles I and III was compared between lactose intolerant vs. lactose tolerant patients and the results are given as mean difference (95% confidence interval). *P*-values <0.05 were considered significant. All tests were carried out as two-sided. Analyses were performed using IBM SPSS Statistics for Windows (version 24.0, IBM Corp., Armonk, NY, USA).

## 3. Results

Patient characteristics are presented in Table 1. The characteristics of the patients and their tumours of those who were included in the FODMAP study (*n* = 52) did not differ significantly from those of the excluded LIPSYT patients in the nutritional intervention study (*n* = 68) (Table 1).

### 3.1. Dietary Intakes

Based on the food diaries, 47 (90%) out of the 52 subjects consumed lactose-rich foods, with their median intake being 3.2 dL/day (range, 0–13.3) at baseline before cycle I and 2.9 dL/day (range, 0–17.3) during cycle III. As expected, lactose intolerant patients (those with hypolactasia or who had a borderline finding in the lactose tolerance test at baseline) consumed less lactose-rich foods during both periods than lactose tolerant patients [when considering both periods simultaneously, mean difference (95% CI) −2.0 dL/day (−3.5 to −0.6), *P* = 0.006]. All 52 subjects had FODMAP-rich foods in their diet, with the median intake being 5.5 portions/day (range, 1.0–13.8) at baseline before cycle I and 5.4 portions/day (range 0–13.8) during cycle III.

### 3.2. Association of Dietary Intakes with Gastrointestinal Symptoms

The patients who consumed lactose-rich foods above the median value were twice as likely to experience diarrhoea during chemo(radio)therapy than those consuming less than the median (Table 2). A dose response was seen when the consumption of lactose-rich foods was divided in three ordered consumption categories (the highest, moderate and the lowest). Patients who consumed the greatest amounts of lactose-rich foods had the highest risk for developing diarrhoea during adjuvant therapy, OR 4.23 (95% CI 1.37 to 13.04, *P* = 0.01) and those who consumed a moderate amount had an intermediate risk, 2.31 (0.75–7.10, *P* = 0.14) as compared with those with the lowest consumption. Flatulence was less frequent among those with a high intake of lactose-rich foods (Table 2).

No statistically significant differences in gastrointestinal symptoms were found in the univariate analysis in patients who consumed FODMAP-rich foods above (High) or below (Low) the median amount (Table 2) nor when the consumption of FODMAP-rich foods was divided into three ordered consumption categories. During the treatment periods, the median intake of FODMAPs was 6.5 portions in the above-median group (High) and 3.9 portions in the below-median (Low) group.

According to the composite variable analysis, patients who consumed both lactose-rich and FODMAP-rich foods above the median (High) amount had a higher risk (OR 4.16) of developing diarrhoea during chemo(radio)therapy in comparison to those whose intake of both lactose-rich and FODMAP-rich foods was below the median (Low) amount (Table 3). Patients who consumed lactose-rich foods above the median (High) but FODMAP-rich foods below the median (Low) had a higher risk (OR 4.18) of developing diarrhoea, but also a significantly reduced risk for developing flatulence during chemo(radio)therapy as compared to those whose intakes of both lactose-rich and FODMAP-rich foods were below the median. Patients who consumed FODMAP-rich foods above the median, but lactose-rich foods below the median, tended to have more than a doubled risk of diarrhoea as compared to those whose intakes of both lactose-rich and FODMAP-rich foods were below the median (Table 3). The background characteristics of the patients, such as the type of surgery or the stage of cancer, did not explain the consumption of FODMAPs or lactose-rich foods; the results are depicted in Appendix A.

### 3.3. Association of Patient Characteristics with Gastrointestinal Symptoms

In addition to the type of food consumed, also other factors were significantly associated with diarrhoea during therapy in the univariable analyses and, therefore, these factors were included in a multivariable analysis. In the final model, diarrhoea was explained by a high consumption of both lactose-rich and FODMAP-rich foods and the presence of lactose intolerance at baseline, whereas if the patient was a methane producer, this was a protective factor (Table 4). According to the final model of flatulence, the protective factors were low consumption of FODMAP-rich foods combined to high consumption of lactose-rich foods and the simplified de Gramont chemotherapy, whereas the presence of lactose intolerance at the baseline increased the likelihood of flatulence. 

## 4. Discussion

We investigated whether consumption of foods rich in poorly absorbed short chain carbohydrates would be associated with gastrointestinal symptoms during adjuvant therapy for colorectal cancer. We found a clinically relevant increase in the risk of diarrhoea in those patients who consumed significant amounts of lactose-rich foods, especially if they also consumed FODMAP-rich foods (multivariate analysis, Table 4).

To the best of our knowledge, no prior studies have assessed the effects of lactose-rich or FODMAP-rich foods on gastrointestinal adverse effects attributable to chemotherapy. In one trial, lactose restriction failed to prevent the diarrhoea induced by whole pelvis irradiation in cancer patients [16]. Similarly, a reduction in the intake of dietary fibre and lactose exerted no effect on the gastrointestinal side effects caused by local radiotherapy in prostate cancer patients [17]. These two findings may not be surprising, since radiotherapy usually causes only local injury in the small intestine, leaving most of the bowel intact.

It has been previously demonstrated that certain chemotherapeutic drugs, such as 5-fluorouracil, increase intestinal permeability in mice [18]. In addition, a systematic review of clinical studies concluded that chemotherapeutic drugs induce major changes in the composition of the gut microbiota and these disruptions can contribute to the development of mucositis, i.e., diarrhoea [19]. It is of interest that when mice were fed a high-FODMAP diet, this not only induced increased osmosis and fermentation in gut, but also a major change in the microbiota of the animals [20]. Furthermore, the level of the pro-inflammatory lipopolysaccharide (LPS) increased in lumen of gut in parallel with the change in the microbial population when the FODMAP intake was high and this level was reduced when FODMAP intake was low [20]. The clinical part of the same study [20] suggested that FODMAPs might also increase faecal LPS levels in humans. Our working hypothesis was that chemotherapeutic drugs cause a loss of intestinal barrier function, disrupt homeostasis of microbiota and disturb enzyme production in the small bowel and thereby sensitize colorectal cancer patients to the osmotic and fermentative effects of FODMAPs and lactose. Furthermore, a high intake of FODMAPs might further worsen the mucositis by increasing the amount of the pro-inflammatory LPS compound in the gut. 

In our previous study, the presence of hypolactasia during adjuvant therapy was associated with both flatulence and diarrhoea [6]. In the present study, however, flatulence was less frequent during adjuvant therapy in patients who consumed above the median levels of lactose-rich foods. This seemingly contradictory finding may indicate that flatulence is more closely linked to hypolactasia during adjuvant therapy, whereas diarrhoea is associated with the consumption of lactose-rich foods. The results from the multivariable models were in line with this hypothesis, i.e., diarrhoea was more strongly associated with the consumption of lactose-rich and FODMAP-rich foods than with presence of lactose intolerance prior to the initiation of the treatments. Flatulence, on the other hand, was more strongly linked with the presence of lactose intolerance at the baseline rather than with the consumption of lactose-rich foods. 

The limitations of the present study are the small patient numbers and that there might be some inaccuracies in the calculation of the intakes of lactose-rich and FODMAP-rich foods. We tried our best to take the latter limitation into account by using categories estimated as decilitres or portions instead of exact grams of intake. In addition, inaccuracies due to the incompleteness of the dietary analysis software were avoided. However, a limitation of this way of assessing the FODMAP content is that the specifics of the diet are lost and there is a risk of either underestimating or overestimating the FODMAP content. For example, the designation of FODMAP foods into our categorization was not comprehensive [21]; those FODMAP foods that tended to be minor parts of usually prepared meals were rather poorly reported as recipes and consequently we were not able to count them reliably from food diaries. However, we are confident that we captured the most relevant FODMAP foods consumed by our participants. Another limitation of the present study in assessing the role of these carbohydrates in causing gastrointestinal symptoms is that the dietary habits may change when the patient’s symptoms worsen. Inter-subject variations in the outcomes were quite large and p-values relatively high. Despite of these limitations, the findings of the present study encourage design and conduction of ten new randomized studies on low FODMAP diet in this particular patient group. Indeed, a randomized multi-centre phase II study led by one researcher from our group is underway where these patients are recommended to consume a low FODMAP diet. 

It is noteworthy that different means of restricting caloric intake may also offer protection from chemotoxicity. Based both on animal experiments and the first human pilot studies, it seems that short-term fasting may reduce the severity of chemotoxicity [22]. However, more robust human trials are urgently needed to clarify the pros and cons of this approach in a clinical setting.

Diarrhoea may be a dose-limiting factor in cytotoxic therapy; it is often the major toxic adverse effect of regimens containing fluoropyrimidines (i.e., 5-fluorouracil) and irinotecan and can be life-threatening [1]. In a randomised trial in colorectal cancer patients, Ravasco et al. demonstrated that an appropriate manipulation of the diet based on regular foods by individual dietary counselling could be beneficial in reducing the gastrointestinal symptoms evoked by radiotherapy [23]. Furthermore, a low-FODMAP diet reduced radiotherapy-induced gastrointestinal symptoms in a non-controlled trial [11]. In line with these data, our results suggest that a low-FODMAP diet may alleviate diarrhoea also during adjuvant chemotherapy for colorectal cancer.

## 5. Conclusions

In conclusion, patients who consumed high amounts of lactose and other FODMAPs had more diarrhoea during adjuvant therapy of colorectal cancer as compared to patients who consumed low amounts of lactose and FODMAPs. A diet low in lactose and FODMAPs may thus be a means to prevent and/or to alleviate the severity of the diarrhoea occurring during adjuvant therapy. This hypothesis warrants further evaluation in a prospective randomised study. 

## Figures and Tables

**Figure 1 nutrients-12-00407-f001:**
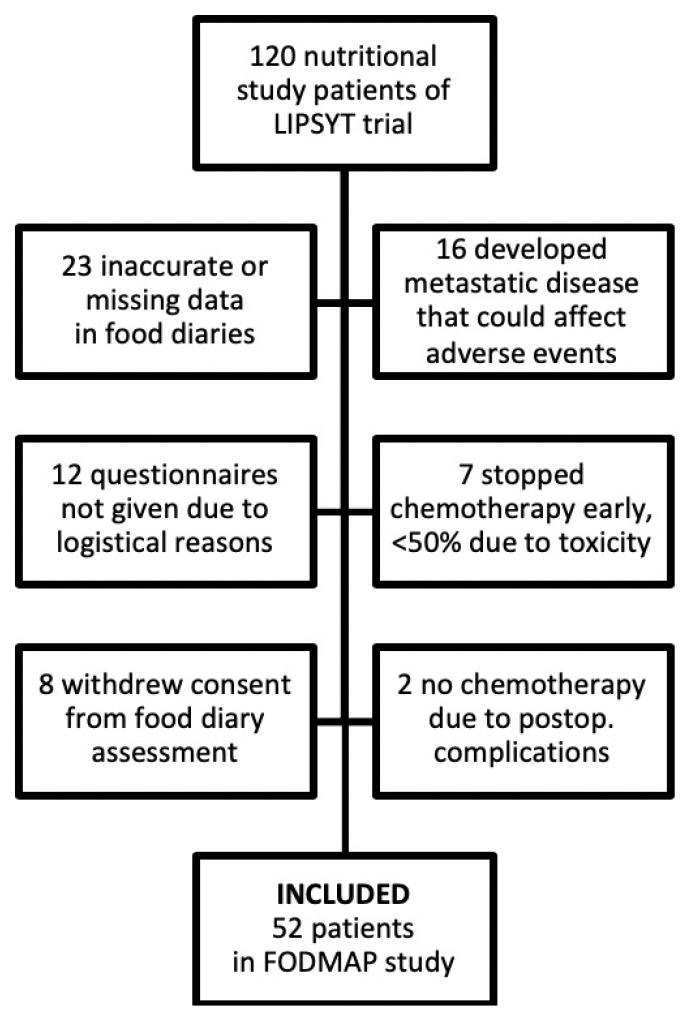
Flowchart showing the 120 patients originally included in the LIPSYT (Liitännäishoito: Paksu- ja PeräsuoliSYöpäTutkimus, Finnish acronym) nutritional intervention study and the 52 patients examined in the FODMAP (fermentable, oligo-, di-, mono-saccharides and polyols) study. Reasons for exclusion are provided.

**Table 1 nutrients-12-00407-t001:** Comparison of the baseline characteristics of the LIPSYT trial patients included in the present study with the patients who were excluded.

Characteristic	Included Patients (*n* = 52)		Excluded Patients (*n* = 68)		
	No. of Patients	%	No. of Patients	%	*P*-Value *
Age (years; median [range])	59	35–74	60	31–76	0.41
Gender					1.00
Male	26	50	35	51	
Female	26	50	33	49	
Lactase deficiency status					0.26
Normolactasia	35	71	49	75	
Borderline	5	10	2	3	
Hypolactasia	9	18	14	22	
Not available	3		3		
Methane producer status					0.58
Producer	18	41	20	36	
Non-producer	26	59	35	64	
Not available	8		13		
Cancer stage					0.44
Stage II	9	17	20	29	
Stage III	36	69	40	59	
Stage IV	7	14	8	12	
Site of cancer					0.76
Colon	32	62	40	59	
Rectum	20	38	28	41	
Type of surgery					0.89
Right hemicolectomy	13	25	13	19	
Left hemicolectomy or sigma Resection	15	29	19	28	
Rectum resection	20	38	29	43	
Subtotal colectomy	4	8	7	10	
Abdominal stoma					0.38
Yes	16	31	16	24	
No	36	69	52	76	
Time since surgery to study entry (weeks; median [range])	5	2.5–14	8	4–16	0.61
Chemotherapy					0.87
Mayo regimen	26	50	35	51	
sLV5FU2 **	26	50	33	49	
Pelvic radiotherapy					0.17
Preoperative short course	6	12	2	3	
Postoperative chemoradiation	14	27	22	32	
None	32	61	44	65	
Type of supplementation					0.82
*Lactobacillus* only	18	34	23	34	
*Lactobacillus* + fibre	15	29	25	37	
No supplements	19	37	20	29	

* Chi^2^ test. Age at study entry and the time since surgery were compared with the Mann-Whitney U test. ** Simplified de Gramont reg.

**Table 2 nutrients-12-00407-t002:** Prevalence of chemo(radio)therapy-related gastrointestinal symptoms of at least mild intensity (Grade 1–4) during cycles I and III according to the consumption of lactose-rich and FODMAP-rich foods in 52 colorectal cancer patients (univariate analysis).

		Diarrhoea *		Constipation		Vomiting		Flatulence		Dyspepsia		Stomatitis	
		*n*	%	*n*	%	*n*	%	*n*	%	*n*	%	*n*	%
Consumption of lactose-rich foods ^†^													
Cycle I	Low (*n* = 26)	6	23	6	23	15	58	12	46	4	15	19	73
	High (*n* = 26)	15	58	6	23	17	65	6	23	7	27	14	54
Cycle III	Low (*n* = 26)	11	42	8	31	16	62	8	31	6	23	11	42
	High (*n* = 26)	14	54	4	15	14	54	4	15	1	4	19	73
High vs. Low ^‡^	OR 95% CI	2.63 1.09–6.37		0.73 0.30–1.78		1.00 0.46–2.15		0.38 0.15–0.95		0.79 0.28–2.22		1.24 0.58–2.64	
	*P*-value	0.03		0.48		1.00		0.04		0.65		0.58	
Consumption of FODMAP-rich foods ^§^													
Cycle I	Low (*n* = 24)	9	38	5	21	14	58	8	33	3	13	16	67
	High (*n* = 28)	12	43	7	25	18	64	10	36	8	29	17	61
Cycle III	Low (*n* = 26)	10	38	8	31	15	58	4	15	4	15	11	42
	High (*n* = 26)	15	58	4	15	15	58	8	31	3	12	19	73
High vs. Low ^‡^	OR 95% CI	1.82 0.72–4.56		0.68 0.29–1.64		1.07 0.60–1.92		1.74 0.67–4.47		1.77 0.59–5.32		1.65 0.66–4.14	
	*P*-value	0.20		0.39		0.81		0.25		0.31		0.28	

* Adverse events were assessed and graded according to the Common Toxicity Criteria of the National Cancer Institute of Canada scale version 2. ^†^ Low = below and High = above median consumption. Cycle I median = 3.2 dL/day, Cycle III median 2.9 dL/day. ^‡^ Logistic regression analysis using generalized estimation equations (GEE)–method. ^§^ Low = below and High = above median consumption. Cycle I median = 5.5 portions/day, Cycle III median 5.4 portions/day. OR = Odds ratio, CI = Confidence interval.

**Table 3 nutrients-12-00407-t003:** Prevalence of chemo(radio)therapy-related gastrointestinal symptoms of at least mild intensity (Grade 1–4) during cycles I and III according to the combinations of lactose-rich and FODMAP-rich foods in 52 colorectal cancer patients (univariate analysis).

Consumption of Foods Rich in *		Diarrhoea ^†^		Constipation		Vomiting		Flatulence		Dyspepsia		Stomatitis	
		*n*	%	*n*	%	*n*	%	*n*	%	*n*	%	*n*	%
Cycle I	None, Lactose Low − FODMAP Low (*n* = 14)	3	21	3	21	7	50	7	50	1	7	10	71
	Lactose High − FODMAP Low (*n* = 10)	6	60	2	20	7	70	1	10	2	20	6	60
	Lactose Low − FODMAP High (*n* = 12)	3	25	3	25	8	67	5	42	3	25	9	75
	Lactose High − FODMAP High (*n* = 16)	9	56	4	25	10	63	5	31	5	31	8	50
Cycle III	None, Lactose Low − FODMAP Low (*n* = 15)	4	27	5	33	10	67	4	27	3	20	6	40
	Lactose High − FODMAP Low (*n* = 11)	6	55	3	27	5	45	0	0	1	9	5	45
	Lactose Low − FODMAP High (*n* = 11)	7	64	3	27	6	55	4	36	3	27	5	45
	Lactose High − FODMAP High (*n* = 15)	8	53	1	7	9	60	4	27	0	0	14	93
Lactose High − FODMAP Low vs. None ^‡^	OR 95% CI *P*-value	4.16 1.25–13.81 0.02		0.89 0.26–3.07 0.85		0.76 0.33–1.77 0.53		0.10 0.02–0.54 0.01		0.93 0.16–5.57 0.94		0.95 0.28–3.25 0.94	
Lactose Low − FODMAP High vs. None ^‡^	OR 95% CI *P*-value	2.59 0.77–8.71 0.13		0.80 0.22–2.86 0.73		0.85 0.32–2.28 0.75		1.35 0.45–3.99 0.59		2.16 0.41–11.21 0.36		1.34 0.40–4.43 0.63	
Lactose High − FODMAP High vs. None ^‡^	OR 95% CI *P*-value	4.18 1.28–13.67 0.02		0.54 0.18–1.59 0.26		1.07 0.49–2.31 0.87		0.71 0.22–2.32 0.57		1.35 0.29–6.38 0.71		1.90 0.65–5.54 0.24	

* Lactose Low, when the consumption of lactose-rich foods was below the median (3.2 dL/day during Cycle I, 2.9 dL/day during Cycle III) and Lactose High, when the consumption was above the median. FODMAP Low, when the consumption of FODMAP-rich foods was below the median (5.5 portions/day during Cycle I, 5.4 portions/day during Cycle III) and FODMAP High, when the consumption was above the median. ^†^ Adverse events were assessed and graded according to the Common Toxicity Criteria of the National Cancer Institute of Canada scale version 2. ^‡^ Logistic regression analysis using generalized estimation equations (GEE)–method.

**Table 4 nutrients-12-00407-t004:** Results of multivariate analysis for chemo(radio)therapy-related diarrhoea and flatulence of at least mild intensity (Grade 1–4) during cycles I and III in 52 colorectal cancer patients.

	Diarrhoea *			Flatulence		
	OR	95% CI	*P*-Value	OR	95% CI	*P*-Value
Consumption of foods						
Lactose High − FODMAP Low ^†^	10.08	0.85–119.2	0.07	0.09	0.01–1.10	0.06
Lactose Low − FODMAP High ^†^	1.73	0.16–18.51	0.65	0.39	0.05–3.11	0.37
Lactose High − FODMAP High ^†^	17.69	1.74–179.7	0.02	0.79	0.14–4.55	0.79
Covariates:						
Chemotherapy regimen (simplified de Gramont vs. Mayo)	-			0.09	0.02–0.51	0.01
*Lactobacillus* GG intervention (yes vs. no)	-			-		
Fibre supplement (yes vs. no)	-			-		
Baseline lactose intolerance (yes vs. no)	4.96	0.88–28.02	0.07	4.03	0.70–23.22	0.12
Presence of stoma (yes vs. no)	-			-		
Methane-producer status (yes vs. no)	0.26	0.05–1.25	0.09	-		
Postoperative chemoradiation (yes vs. no)	-			-		

* Adverse events were assessed and graded according to the Common Toxicity Criteria of the National Cancer Institute of Canada scale version 2. ^†^ Lactose Low, when the consumption of lactose-rich foods was below the median (3.2 dL/day during Cycle I, 2.9 dL/day during Cycle III) and Lactose High, when the consumption was above the median. FODMAP Low, when the consumption of FODMAP-rich foods was below the median (5.5 portions/day during Cycle I, 5.4 portions/day during Cycle III) and FODMAP High, when the consumption was above the median.

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
