# Peer review of "Consumption of Lactose, Other FODMAPs and Diarrhoea during Adjuvant 5-Fluorouracil Chemotherapy for Colorectal Cancer"

_nutrients, 2020, doi:10.3390/nu12020407_

Round 1
Reviewer 1 Report
Please consider an acknowledgement in the discussion that there are large inter-subject variations in the measured outcomes of this study and that the p-values obtained are therefore of rather low statistical significance. I do not suggest that the results and conclusions are not essentially correct, but the inherent uncertainties do indicate the need for further, larger studies to confirm and develop any changes that may be required in clinical practice.
Author Response
"Please consider an acknowledgement in the discussion that there are large inter-subject variations in the measured outcomes of this study and that the p-values obtained are therefore of rather low statistical significance. I do not suggest that the results and conclusions are not essentially correct, but the inherent uncertainties do indicate the need for further, larger studies to confirm and develop any changes that may be required in clinical practice."
We have updated the limitations, lines 325-329 as suggested by the reviewer:
“Inter-subject variations in the outcomes were relatively large and p-values relatively high. Despite of these limitations, the findings of the present study support prospective interventional studies where the intake of FODMAPs is reduced in a patient population receiving cancer chemotherapy frequently associated with diarrhoea. Indeed, a randomized multicenter phase II study onlow FODMAP diet in this patient group is underway and led by one senior researcher from our group.”
We are indeed happy to inform that there is a randomized multicenter phase II study ongoing, led by our senior researcher Pia Österlund, further investigating the role of FODMAPs in this patient group.
Reviewer 2 Report
Holma H. et al describe a correlation between the FODMAPS and lactose consumption and symptoms during chemotherapy. The issue is vey interesting for any reader and the manuscript, with all the limits of a retrospective analysis, is sound.
There are few minor points that this referee would like to be addressed:
1) Is it possible to have an account also of the caloric intake of the patients?
2) is 5 fluorouracile a very frequent adjuvant in Colon Carcinoma? What is the frequency of use and a what stage of the disease?
3) A comment, in the discussion, on the effect of caloric restriction on the chemotherapy symptoms is needed (I.E. Fasting vs dietary restriction in cellular protection and cancer treatment: from model organisms to patients.
Lee C, Longo VD.Oncogene. 2011 Jul 28;30(30):3305-16. doi: 10.1038/onc.2011.91. Epub 2011 Apr 25. Review.)
4) were other symptoms, a part from gastrointestinal, related to chemotherapy analysed? Did they improve or not?
Author Response
Holma H. et al describe a correlation between the FODMAPS and lactose consumption and symptoms during chemotherapy. The issue is very interesting for any reader and the manuscript, with all the limits of a retrospective analysis, is sound.
There are few minor points that this referee would like to be addressed:
1) Is it possible to have an account also of the caloric intake of the patients?
There is no caloric intake calculated for the patients in this study, unfortunately.
2) is 5 fluorouracile a very frequent adjuvant in Colon Carcinoma? What is the frequency of use and a what stage of the disease?
5-fluorouracil is the cornerstone of treatment for gastrointestinal cancers, especially colorectal cancer both in the adjuvant and in the metastatic setting. We have added this information into the text, lines 71-2:
“The extent to which lactose and other poorly absorbed short chain carbohydrates contribute to gastrointestinal adverse effects caused by 5-fluorouracil, the cornerstone for gastrointestinal cancer treatments in both the adjuvant and metastatic setting, or other chemotherapy agents is unknown.”
Labianca R, Nordlinger B, Beretta GD, Mosconi S, Mandala M, Cervantes A, et al. Early colon cancer: ESMO Clinical Practice Guidelines for diagnosis, treatment and follow-up. Ann Oncol. 2013;24 Suppl 6:vi64-72. van Cutsem: ESMO consensus guidelines for the management of patients with metastatic colorectal cancer. Ann Oncol. 2016;27(8):1386-422.3) A comment, in the discussion, on the effect of caloric restriction on the chemotherapy symptoms is needed (I.E. Fasting vs dietary restriction in cellular protection and cancer treatment: from model organisms to patients.)
Lee C, Longo VD.Oncogene. 2011 Jul 28;30(30):3305-16. doi: 10.1038/onc.2011.91. Epub 2011 Apr 25. Review.)
This is indeed interesting and rapidly developing area. We have added the following sentence into the latter part of the discussion, lines 331-333.
“It is noteworthy that different means of restricting caloric intake may also offer protection from chemotoxicity. Based on animal and first human pilot studies, short-term fasting seems to reduce chemotoxicity [22]. However, more robust human trials are urgently needed to understand pros and cons of such methods in clinical setting.”
4) were other symptoms, a part from gastrointestinal, related to chemotherapy analysed? Did they improve or not?
In this FODMAP-study we focused only on gastrointestinal symptoms, as low FODMAP diet is mainly considered as a treatment for gastrointestinal symptoms. The other toxicities related to the drug treatment have been rigorously explored with patient diaries etc. and previously reported [references 12 and 6 in the current paper]
Reviewer 3 Report
The manuscript of Holma et al. reported a clinical association test between consumption of lactose and other fermentable sugars and gastrointestinal symptoms, in the colorectal cancer patients received adjuvant 5-fluorouracil chemotherapy. There is not sufficient evidence in oncological patients to consider a low FODMAP diet in the prevention or in the treatment of cancer-related diarrhea. Therefore the manuscript could be very beneficial this direction. However revision is suggested to solidify the manuscript.
Major comments,
As discussed in the introduction, the relevance of fermentable sugar associated GI symptom and chemotherapy caused GI damage is straightforward. The higher amount of fermentable sugar consumption will cause more severe GI symptoms. It is interesting to understand the efficacy of lactose restriction diet in reducing or preventing GI issues during chemotherapy. Why the author did not include FODMAP-restriction diet in the study?
The level of lactose and FODMAPs intake is very important in the study design. Although the limitation has been discussed in the manuscript, it would be helpful if authors add a supplementary table to present the individual data of sugar consumption under each categories of Lactose/FODMAP Low, Moderate, and High.
The synergistic effect of Lactose and FODMAP diet on diarrhea is highlighted in the manuscript. However, it is not clear in the results (table 3). The addition of FODMAP-High diet (under Lactose-High condition) was not different with FODMAP-Low in causing diarrhea.
Minor comments,
Line 237-238. Please discuss whether it is appropriate to interpret “OR=4.18” as “a 4-fold risk”
Line 317-324. It is confusing discussion. Please rewrite.
Author Response
The manuscript of Holma et al. reported a clinical association test between consumption of lactose and other fermentable sugars and gastrointestinal symptoms, in the colorectal cancer patients received adjuvant 5-fluorouracil chemotherapy. There is not sufficient evidence in oncological patients to consider a low FODMAP diet in the prevention or in the treatment of cancer-related diarrhea. Therefore, the manuscript could be very beneficial this direction. However, revision is suggested to solidify the manuscript.
Major comments,
As discussed in the introduction, the relevance of fermentable sugar associated GI symptom and chemotherapy caused GI damage is straightforward. The higher amount of fermentable sugar consumption will cause more severe GI symptoms. It is interesting to understand the efficacy of lactose restriction diet in reducing or preventing GI issues during chemotherapy. Why the author did not include FODMAP-restriction diet in the study?
We are not sure if we understand the focal point of the question raised; our study is a retrospective analysis of previously published RCT focusing on Lactobacillus GG (12).
When the pivotal study (12) was initiated, primary interest was the effect of Lactobacillus GG on the GI symptoms of colorectal cancer patients. The whole concept of FODMAPs was unknown at that time (2007), FODMAP concept has only emerged during the 10 years or so. In other words, our current study is a post hoc analysis of a previously randomized study in which food consumption was monitored with food diaries, thus allowing a post-intervention estimation of FODMAP intake in a naturalistic setting, as the FODMAP concept emerged with time.
We think that our current study may serve as a major impetus for future randomized FODMAP-restriction studies in this patient group – due to its inherent weakness (retrospective, observational study) our study cannot be perceived as a sole foundation for recommending low FODMAP diet for this patient group.
Actually, there is now a randomized multicenter phase II study ongoing, led by our senior researcher Pia Österlund, further investigating the particular question.
The level of lactose and FODMAPs intake is very important in the study design. Although the limitation has been discussed in the manuscript, it would be helpful if authors add a supplementary table to present the individual data of sugar consumption under each categories of Lactose/FODMAP Low, Moderate, and High.
We acknowledge the need for best possible evaluation of FODMAP intake in the context of clinical studies.
As described in the text (lines 190-193, paragraph 2.5.), we used categories low vs high in all our analyses presented in the tables –not terciles (low, moderate, high) even if they are mentioned in the methods, and in the results.
We have used FODMAP servings (as described by Monash team, J Gastroenterol Hepatol. 2017;32 Suppl 1:53-61, J Hum Nutr Diet. 2011 Apr;24(2):154-76) rather than reported intake of individual FODMAPs as a basis for comparisons (high vs low) because there is no database/software on individual FODMAP content of Finnish food items. Individual FODMAP categories (fructans, GOS, polyols, lactose and excess fructose) would indeed be scientifically interesting, but unfortunately, we are not able to do analyze them due to the lack of relevant database.
For these two reasons, we consider lines 236-8. “The median intake of FODMAPs was 6.5 portions in the above-median (High) group and 3.9 portions in the below-median (Low) group the during the treatment periods.” as satisfying enough considering the retrospective, hypothesis generating nature of the current study.
The synergistic effect of Lactose and FODMAP diet on diarrhea is highlighted in the manuscript. However, it is not clear in the results (table 3). The addition of FODMAP-High diet (under Lactose-High condition) was not different with FODMAP-Low in causing diarrhea.
We agree that in the univariate analysis (table 3) there does not seem to be difference between the two conditions (ORs are 4.16 and 4.18). However, table 4 illustrates a more robust multivariate analysis with seven relevant covariates. As far as we understand, univariate analysis (table 3) emphasizes description of the data while multivariate methods emphasize hypothesis testing and explanation (table 4). In our multivariate analysis, table 4, only Lactose High – FODMAP highdiet was associated with statistically significantly increased risk of diarrhea (OR 17.69, P=0.02); Lactose High – FODMAP low diet’s OR was 10.08, P=0.07. The very relevance of the multivariate analysis is also highlighted by the fact that pure lactose intake has not been not associated with diarrhea in previous studies (references 16,17).
Therefore, we think that synergistic effect is truthfully presented in the text. However, as we have now become aware of the source of potential confusion, we now refer to table 4 in the revised text to underline where the synergistic effect was found. Revised sentence, lines 284-5, now writes: “We found a clinically relevant increase in the risk of diarrhoea in those who consumed abundantly lactose-rich foods, especially if they consumed also FODMAP-rich foods; multivariate analysis table 4."
Minor comments,
Line 237-238. Please discuss whether it is appropriate to interpret “OR=4.18” as “a 4-fold risk”
The text is now revised to better illustrate true meaning of the observed ORs.
Line 317-324. It is confusing discussion. Please rewrite.
We agree that this part is somewhat confusion, and we have now re-considered the whole value of this discussion on lactose. After careful consideration, we conclude that this part of the text is better to be omitted as the issue is complex, very difficult to discuss clearly and concisely, and of minor importance as whole. Therefore, we have deleted the confusing lactose discussion.
Round 2
Reviewer 3 Report
The authors have addressed review comments sufficiently. The revised manuscript is stronger now.
Author Response
Thank you for your comments and review.
Language has now gone through extensive editing.
